# Fabrication and Performance of Self-Supported Flexible Cellulose Nanofibrils/Reduced Graphene Oxide Supercapacitor Electrode Materials

**DOI:** 10.3390/molecules25122793

**Published:** 2020-06-17

**Authors:** Wen He, Bo Wu, Mengting Lu, Ze Li, Han Qiang

**Affiliations:** College of Materials Science and Engineering, Nanjing Forestry University, Nanjing 210037, China; umengting12345612@163.com (M.L.); lze0125@163.com (Z.L.); qianghan1016@163.com (H.Q.)

**Keywords:** cellulose nanofibrils, reduced graphene oxide, supercapacitor, electrode

## Abstract

With the rapid development of portable and wearable electronic devices, self-supporting flexible supercapacitors have attracted much attention, and higher requirements have been put forward for the electrode of the device, that is, it is necessary to have good mechanical properties while satisfying excellent electrochemical performance. In this work, a facile method was invented to obtain excellent self-supported flexible electrode materials with high mechanical properties and outstanding electrochemical performance by combining cellulose nanofibrils (CNFs) and reduced graphene oxide (RGO). We focused on the effect of the ratio of the addition of CNFs and the formation process of the film on the electrochemical and mechanical properties. The results show that the CNFs/RGO_12_ (where the ratio of CNFs to GO is 1:2) film displayed outstanding comprehensive properties; its tensile strength and conductivity were up to 83 MPa and 202.94 S/m, respectively, and its C_A_ value was as high as 146 mF cm^−2^ under the current density of 5 mA cm^−2^. Furthermore, the initial retention rate of the specific capacitance was about 83.7% when recycled 2000 times; moreover, its capacitance did not change much after perpendicular bending 200 times. Therefore, the films prepared by this study have great potential in the field of flexible supercapacitors.

## 1. Introduction

As a result of the great shortage of nonrenewable energy resources, green recyclable energy has become increasingly important around the world. High-performance energy storage devices can, to a large extent, expand the application of renewable energy and decrease the energy loss; therefore, research on green energy storage devices has received additional consideration in energy storage fields [1,2]. Supercapacitors have become a topic of wide interest because of their rapid charge–discharge rate, high power density, long service life, high safety, and so on [3,4,5].

Recently, a variety of supercapacitor electrodes have been developed mainly using carbon-based materials, conductive polymers, and metallic oxide [6,7,8,9,10,11,12,13]. Among the above materials, graphene has received considerable attention as a novel carbon material to manufacture flexible electrodes by virtue of its high specific area, high conductivity, excellent mechanical properties, and prominent chemical stability [14,15,16]. However, the ineluctable restacking and strong van der Waals force of graphene nanosheets result in a lower electrochemical performance, which restricts their application [17,18,19]. To solve these problems, graphene is usually combined with conductive polymers, metallic oxide, or cellulose nanofibers [17,20,21,22,23]. It is noteworthy that cellulose nanofibers have already displayed unique superiority when combined with graphene to fabricate flexible supercapacitor electrodes [24,25,26].

Nanocelluloses can be classified into three main types, namely, cellulose nanocrystal, cellulose nanofibers, and bacterial nanocellulose, based on their source and dimensional differences [27,28]. Among them, cellulose nanofibers have gained wide attention because of their high tensile strength, huge aspect ratio, rich carboxyl content, and so on. They are usually extracted from biomass fibers by the combination of 2,2,6,6-tetramethylpiperidin–1-oxyl radical (TEMPO) oxidizing and ultrasonic processing [29,30]. In this process, TEMPO oxides the hydroxyl located in C6 of the cellulose molecule to carboxyl, and the negative charges on the carboxyl are of mutual repulsion, which weakens intermolecular hydrogen bonding, leading to the microfibrils being easily isolated into cellulose nanofibers only under moderate ultrasound treatment [31,32]. As a result, graphene oxide (GO) can be uniformly dispersed in the cellulose nanofibril (CNF) solution under the action of rich carboxyls and hydroxyls [33]. After the films have been fabricated, the CNFs embed into the reduced graphene oxide (RGO) nanosheets to restrain their accumulation. In addition, the hydrophilicity, porosity, and hydroexpansivity of CNFs may be propitious to offer enough diffusion channels for the electrolyte solution and to enhance the contact between electrodes and electrolytes, which may reduce the internal resistance and charge transfer obstruction [34,35,36,37]. In return, the electrochemical performance of the RGO is greatly improved.

At present, CNF/RGO composites are continually prepared to apply to supercapacitor electrodes. Their fabrication is mainly classified into two types. One is to mix CNFs and GO in distilled water, and then directly filtrate the solution into paper-thin films under vacuum pressure. In this process, the CNF/RGO composite films develop a compact structure, which results in a higher tensile strength; however, this structure is not in favor of the transmission of electrolytes. The other fabrication method is to forthright freeze-dry the CNF/GO suspension in aerogel, and then compress it into films using a press. The obtained films have an abundant multiscale pore structure, which is better for the flow of electrolytes and charge transferring; nevertheless, the mechanical properties are not ideal due to the loose connection between the CNFs and RGO [38,39,40,41]. However, excellent self-supporting flexible electrode materials require higher mechanical properties and capacitance retention, therefore, it is necessary to deal with the problem of bonding between materials.

In this work, a facile method was invented to obtain excellent self-supported flexible electrode materials with high mechanical properties and outstanding electrochemical performance by combining CNFs and RGO. The CNFs were isolated from bamboo material using a combination of TEMPO oxidation and ultrasonic processing. Then, the GO was dispersed into the CNF suspensions, and subsequently, the obtained mixture was filtered to a gelatinous thickness of 5 mm under vacuum pressure, and then it was placed in an acidic atmosphere to form a more complete hydrogel, finally, the hydrogel was directly desiccated and then reduced into CNF/RGO films. The main effort was focused on the effect of the proportion of CNFs to GO on the microstructure, mechanical properties, conductivity, and electrochemical performance of CNF/RGO composite films.

## 2. Experiment and Methodology

### 2.1. Materials and Chemicals

Bamboo powder (60~100 meshes) with a moisture of 6~8% was collected from Anji County in Zhejiang province, China. NaClO_2_ (95~98 wt%), NaOH (96 wt%), 2,2,6,6-tetramethylpiperidin–1-oxyl radical (TEMPO), NaBr, NaClO (11 wt%; available chlorine ≥5.2%), CH_3_CH_2_OH (98wt%), HCl (98 wt%), and acetic acid (36 wt%) were analytical reagents and purchased from Sinopharm Chemical Reagent Co., Ltd. Hydriodic acid (45~55 wt%) and absolute ethyl alcohol (99 wt%) were obtained from Shanghai Macklin Biochemical Technology Co., Ltd. Single-layer GO aqueous dispersion at a concentration of 10 mg/g was purchased from Hangzhou GaoxiTech Co., Ltd.

### 2.2. Preparation of the CNF Suspension

Bamboo powder was put into an NaClO solution (4.5 pH, 2.5 wt%) and heated for 1 h at 80 °C. Then, it was washed with deionized water by vacuum filtration, and the whole process was repeated 6 times. Subsequently, the treated powder was heated in an NaOH solution (5 wt%) at 85 °C for 2 h under stirring, and then washed repeatedly until the solution became neutral. Finally, the solution was vacuum-dried at 100 °C for 24 h to obtain α-cellulose. TEMPO (0.01 g) and NaBr (0.1 g) were dissolved in 100 mL deionized water, before α-cellulose (1 g) was added to the above solution under continuous stirring. Then, 16 mmol of sodium hypochlorite was dripped slowly into the mixed solution, in which the pH was adjusted to approximately 10 with dilute HCl and NaOH solutions, respectively. After reacting for 4~6 h, 10 mL anhydrous ethanol was used to terminate the reaction. Then, the solution was washed repeatedly until the pH was approximately 7. The obtained suspension of 2 mg/mL was ultrasonically treated for 30 min at 600 W power with an ultrasonic processor (JY96-IIN, SCIENTZ, China), before being centrifuged for 10 min at 10,000 r/min by a high-speed refrigerated centrifuge (TGL-16M, BIORIDGE, China). Finally, the supernatant from the centrifugal tube was collected and stored at approximately 4 °C in a refrigerator.

### 2.3. Fabrication of the CNF/RGO Composite Films

The CNF and GO suspensions at a concentration of 1 mg/mL were prepared, and then mixed at a mass ratio of 2:1, 1:1, 1:2, and 1:3. These mixtures were uniformly dispersed by ultrasonic processing for 5 min at 450 W power and marked as CNFs/GO_21_, CNFs/GO_11_, CNFs/GO_12_, and CNFs/GO_13_, respectively.

Each mixture of 35 mL with different proportions was filtered to a thickness of 5 mm by a vacuum filter and sealed in a hydrochloric acid atmosphere for 6–8 h to form a complete gel state. Subsequently, these hydrogels were purged with distilled water and desiccated at 50 °C for 4 h using a vacuum drying oven. Finally, all CNF/GO films were reduced for 6 min at 60 °C using a HI of 50 mL, named as CNFs/RGO_21_, CNFs/RGO_11_, CNFs/RGO_12_, and CNFs/RGO_13_, respectively. Meanwhile, the RGO film was also prepared under the same conditions as a reference.

### 2.4. Characterizations

The CNF film, GO film, RGO film and CNF/GO film, as well as the CNF/RGO composite films, were investigated by Fourier transform infrared spectroscopy (VERTEX80V, Bruke, Germany). The tested spectrum was in the range of 750~4000 cm^−1^ and the resolution ratio was 4 cm^−1^. The Raman spectra of the RGO film and the CNF/RGO composite films were detected with a Raman spectrometer (DXR532, Thermo Fisher, Waltham, MA, USA). The excitation wavelength was 532 nm and the spectrum range was between 500 and 3300 cm^−1^. All of the prepared films were investigated by an X-ray diffractometer (D5005, Munich, Germany) with a scanning rate of 5º/min in the range of 5º~50º, in which the radiographic source was Cu Kα and the wavelength was λ ≈ 1.54 Å. All of the prepared films were checked by X-ray photoelectron spectroscopy (AXIS UltraDLD, Shimadzu, Japan) at a power of 200 W using a spot diameter of 650 um, in which the excitation source was AlKcL. The RGO film and the CNF/RGO composite films were observed using a field emission scanning electron microscope (S4800F, JEOL, Tokyo, Japan) at a designed voltage of 5.0 kV. Specimens of the CNF film, RGO film, and all CNF/RGO composite films with a dimension of 30 × 5 mm (length × width) were tested by a normal mart electron tensile testing machine (SLBL-500N, SHIMADZU, Kyoto, Japan) using a stretching rate of 1 mm/min. Each group specimen was tested 6 times and the averages were recorded. The RGO film and the CNF/RGO composite films were cut into specimens of 20 × 10 mm (length × width), and their conductivity was detected by an electrochemical workstation (CHI 760E, CHI Instruments, Shanghai, China) under a double-electrode system. The calculation formula of conductivity is as follows:(1)R=UI
(2)σ=LRS
where *U* represents the open circuit potential (V), *I* represents the corresponding current (A), *σ* represents the conductivity (S/m), *R* represents the resistance (Ω), *L* is the distance between two electrodes (m), and *S* is the cross-sectional area of the electrode region.

The electrochemical performance of the RGO film and the CNF/RGO composite films was measured by an electrochemical workstation (CHI 760E, CHI Instruments, Shanghai, China) using a three-electrode system in an electrolyte of 0.1 M H_2_SO_4_. The films directly used as the working electrode, the platinum electrode, and the saturated calomel electrode (SCE) were used as the counter electrode and the reference electrode, respectively. Cyclic voltammetry (CV) was performed at 5, 10, 20, 30, 50, and 100 mV s^−1^ scan rates in a potential window of 0–1.0 V. The galvanostatic charge/discharge (GCD) test was carried out in a potential range from 0 to 1.0 V with an applied current density of 0.5, 1.0, 2.0, 3.0, 5.0, 8.0, and 10.0 mA cm^−2^. The electrochemical impedance spectroscopy (EIS) measurement was performed at frequencies ranging from 10 mHz to 100 kHz, with the amplitude set to 10 mV. The areal capacitance (C_A_; mF cm^−2^) was calculated from the GCD curves according to Equation (3):(3)$$C_{A}=\frac{IT}{AU}$$
where *I* (mA), *T* (s), and *U* (V) are the charge/discharge current, the discharge time, and the potential drop during discharge in V, respectively, and *A* is the area of the electrode in cm^2^.

## 3. Results and Analysis

### 3.1. Microstructure and Chemical Characterization

Figure 1 illustrates the preparation process of the CNF/RGO composite films. Previous research indicated that it is easier to isolate the CNFs from bamboo materials than other bioresources due to their fewer transverse tissues; also, these CNFs have a higher depth–diameter aspect ratio than those obtained from wood materials, which is better for dispersing carbon-based nanomaterials [31,32,42,43]. Moreover, in this study, we developed a simple method to fabricate the CNF/RGO films, in which the mixture of CNFs and GO was filtrated to form a film with a higher thickness, and then gelled into hydrogel, before finally being dried into a film. This technology allowed the CNF/RGO film to not only keep its higher mechanical properties, but also to form lots of space with a multiscale pore structure inside and wrinkles on the surface [44].

The microstructure of the films of RGO, CNFs/RGO_13_, CNFs/RGO_12_, CNFs/RGO_11_, and CNFs/RGO_21_ can be observed in Figure 2. A significant laminated structure is shown on the cross-section of all films. In addition, with the increase of the proportion of CNFs, the compact laminated structure of the CNF/RGO films gradually became loose. The compact structure of the RGO film is attributed to the Van der Waals force of the GO nanosheets and the graphene interplanar π–π interaction during reduction; however, inserting CNFs restrained the ordered restacking of the GO nanosheets and weakened the graphene interplanar π–π interaction, which resulted in a loose structure for the CNF/RGO films [45,46]. At the same time, a lot of gullies and wrinkles became visible on the surface of the CNF/RGO composite films. This phenomenon is ascribed to the syneresis that happened in the processes of the reduction and the low-temperature drying of the CNF/RGO films. These distinctive appearances greatly increased the specific area of the CNF/RGO composite films, which is beneficial for use as the electrode materials [47,48].

The FT-IR spectra of the CNF, GO, RGO, CNFs/GO_12_, and CNFs/RGO_12_ films are displayed in Figure 3. The CNF film shows main characteristic peaks located at 3340 cm^−1^, 2895 cm^−1^, 1601 cm^−1^, and 1020 cm^−1^ on its FT-IR spectra curve, which are attributed to the stretching vibration of the –OH, C–H, C–O, and C–O–C groups, respectively. For the FT-IR spectra of the GO film, four characteristic peaks can be observed at 3441 cm^−1^, 1720 cm^−1^, 1637 cm^−1^, and 1385 cm^−1^. These peaks mainly correspond to the –OH, C=O, C=C, and C–OH groups, respectively. Compared to the FT-IR spectra of the GO film, the intensity of the characteristic peaks significantly decreased in the RGO spectra and the peak located at 1385 cm^−1^ nearly disappeared, which proves that the GO was completely reduced. The peak that appeared at 1020 cm^−1^ of the FT-IR spectra curve of the CNF/GO films is mainly ascribed to the stretching vibration of the C–O–C groups of the CNFs, which confirms the successful addition of CNFs. Comparing the CNF/GO and the CNF/RGO films, the main characteristic peaks display similar changes to the results of the GO and the RGO films. This fact proves that the CNF/GO films were successfully reduced into CNF/RGO films.

To further prove the reduction results of the CNF/GO films, the GO, RGO, CNF/GO and CNF/RGO films were investigated by XPS. As shown in Figure 4a, the C1s bonding energy of the GO film shows main characteristic peaks at 284.5 eV, 286.5 eV, 287.8 eV, and 289.0 eV. These characteristic peaks correspond to C–C/C=C, C–O, C=O, and O–C=O of the constitutional formula of GO, respectively. After having been reduced, the C1s bonding energy of RGO shows similar characteristic peaks at the same location, yet the intensity of these peaks is significantly decreased, especially for the C–O groups located at 286.5 eV, as shown in Figure 4b. The C1s bonding energy of the CNF/GO and CNF/RGO films are displayed in Figure 4c,d, respectively. Similar characteristic peaks appear at the same bonding energy location for both of them, and the intensity of these peaks is also greatly decreased for the CNF/RGO films, which confirms that the GO in the CNF/GO films was completely reduced. However, the intensity of the characteristic peaks of C–O, C=O, and O–C=O is increased to varying degrees for the CNF/GO and CNF/RGO films compared to those of the GO and RGO films, which is attributed to the carboxyl group and the other oxygenated groups in the CNFs.

Figure 5a shows the Raman spectra of the GO and the RGO films, in which two characteristic peaks (marked D and G) can be observed. The D peak is usually regarded as the disordered structure region of graphene nanosheets, while the G peak reflects the number of ordered structure regions, which is attributed to the in-plane stretching vibration of the sp^2^ carbon atoms of graphene [49,50]. The ratio of ID/IG is typically used to characterize the degree of defect of graphene nanosheets. After having been reduced, the intensity of the D peak is greatly increased, and the ratio of ID/IG is raised from 1.01 to 1.71 for the RGO film. This change confirmed that the partial restoration of the graphene lattice structure occurred after reduction [51]. The Raman spectra of the RGO and all CNF/RGO composite films are shown in Figure 5b. The D and G peaks in all CNF/RGO spectra gradually shift toward the left with the increase of CNFs, and their ratios of ID/IG are all less than that of RGO, especially for the CNFs/RGO_12_, as its ratio dropped to 1.418. These changes clearly indicate that the interaction between CNFs and RGO is able to develop a new ordered sp^2^ graphite region.

Figure 5c,d displays the X-ray diffraction (XRD) patterns of the CNF, GO, RGO, CNFs/GO_12_, and CNF/RGO films. The main diffraction peaks located at 16.1° and 22.9° in the CNF spectra correspond to the *I*_101_ and *I*_200_ diffraction plane of the classical crystal structure of cellulose I. A clear diffraction peak can be observed at 10.3° in the GO spectra, which is attributed to *I*_002_ plane of graphene. Noticeably, it transformed to the wider diffraction peak located at 24.24° in the RGO pattern. This is because the graphene nanosheets restacked under the effect of π–π bonds after the massive oxygen-containing functional groups were removed during the reduction of GO. On the other hand, the diffraction peak of GO in the CNF/GO films shifts parallel to the lower wavenumber due to the addition of CNFs; similarly, as shown in Figure 5d, the diffraction peak of RGO in the CNF/RGO films gradually shifts to the left with the increase of CNFs. These facts are completely consistent with the above investigation of using FT-IR and XRD. The interlamellar spacing of the RGO, CNFs/RGO_13_, CNFs/RGO_12_, CNFs/RGO_11_, and CNFs/RGO_21_ films was calculated according to Bragg’s formula, and their values are 3.66 Å, 3.69 Å, 3.72 Å, 3.79 Å, and 3.87 Å, respectively. The increase of the interlamellar spacing further proves the fact that the restacking of graphene nanosheets is impeded by the interaction of CNFs as an insert.

### 3.2. Mechanical Properties and Conductivity of the CNF/RGO Composite Films

The stress–strain curves of the RGO and CNF/RGO films are exhibited in Figure 6a. A common view is that CNFs weaken the stack effect of graphene nanosheets and destroy their compact structure [32,52]; however, the addition of CNFs in this study increased the mechanical properties of the RGO film. This change is mainly attributed to the high tensile strength of CNFs; moreover, a great number of carboxyl groups on the CNFs can combine the carboxyls of GO to form a stable interface. It is noteworthy that the tensile strength of all of the CNF/RGO composite films is over 50 MPa, and with a larger elongation at the break, which are significantly higher than that of the CNF/RGO films in some studies [22,53,54]. This enhanced tensile strength is mainly due to the early vacuum filtration in which the compact three-dimensional network structure composed of CNFs and GO was gradually developed with decreasing amounts of water, and then the gelation further strengthened the network of the films due to the formation of the intramolecular and intermolecular hydrogen bonds of CNFs [42]. Moreover, the CNFs/RGO_12_ film showed the highest mechanical properties of all of the composite films in this study, with its tensile strength and strain being around 83 MPa and 5.7%, respectively.

As an outstanding conductive material, graphene can offer excellent conductivity and stable capacitance when used for the electrodes of supercapacitors. However, its compact structure resulting from the gather of nanosheets decreases its electrochemical performance to a large extent. In this study, the conductivity of the CNF/RGO composite films with different CNF contents was tested, and their conductivity was greatly enhanced firstly, before dropping with the continued increase of CNFs, as shown in Figure 6b. The conductivity of the CNFs/RGO_13_ film displayed the highest value, up to 1342.4 S/m, which far exceeds the conductivity of the pure RGO film, whose value is about 934.5 S/m. However, with the increase of CNFs, the conductivity of the CNFs/RGO_12_ film can still reach up to 202.9 S/m.

### 3.3. Electrochemical Performance of the CNF/RGO Composite Films

In order to evaluate the electrochemical performance of the CNF/RGO composite films as electrodes, their CV, GCD, and EIS were measured. The GCD curves of the RGO and CNF/RGO film electrodes at 1 mA cm^−2^ are shown in Figure 7a. All curves present an approximately representative symmetric shape, and all CNF/RGO films display longer charge–discharge times than the RGO film. The C_A_ of all of the electrodes under different current densities are shown in Figure 7b. The RGO film shows the lowest values, and they are caused by its restacking of the nanosheets during reduction. The rise in C_A_ for all of the CNF/RGO films is because the insertion of CNFs provides a great number of electrolyte channels; on the other hand, the water expansion effect of hydrophilic CNFs may absorb a part of the electrolyte, which further leads to a higher C_A_. The C_A_ values of all of the CNF/RGO film electrodes reach over 79 mF cm^−2^ when the current density is increased to 5 mA cm^−2^, and the CNFs/RGO_12_ film exhibits the biggest C_A_ value, approximately 146 mF cm^−2^, which is much higher than those of the CNF/RGO films fabricated by other processing technologies [50,55].

The electric resistance and ion transferring behavior of the CNF/RGO films are characterized by a Nyquist diagram measured under an open circuit potential, which is shown in Figure 7c. The representative impedance curves take on a semicircle in a high-frequency range, as shown in Figure 7c. The intercept of the semicircle at the real axis represents the internal resistance (R_s_), which is mainly dependent on the intrinsic electrical resistance of the electrode material, in addition to the contact and ionic resistance of the electrolyte. Additionally, the radius of the semicircle represents the charge transfer resistance (R_ct_), which mainly reflects the electronic conduction and diffusion abilities of ions [56,57,58,59]. Herein, the R_s_ value of the RGO, CNFs/RGO_13_, CNFs/RGO_12_, CNFs/RGO_11_, and CNFs/RGO_21_ films is 2.3 Ω, 2.6 Ω, 2.8 Ω, 3.2 Ω, and 3.8 Ω, respectively. These R_s_ values show a smaller growth with the addition of CNFs. Yet, the R_ct_ value of the CNF/RGO films is significantly decreased due to the existence of CNFs. The R_ct_ value of the CNFs/RGO_12_ film is around 3.1 Ω, which is reduced by 4.7 Ω compared to that of the RGO film. The decrease of the R_ct_ value is due the hydrophilic CNFs strengthening the interface contact between the electrolyte and the CNF/RGO films. On the other hand, an ideal capacitor would show a vertically straight line at low frequencies, and its rake ratio reflects the capacitive behavior of the electrode. It can be clearly observed that all CNF/RGO films show higher capacitive behavior than that of the RGO film, and the CNFs/RGO_12_ film exhibits the best capacitive behavior of all electrodes.

The CNFs/RGO_12_ film exhibits excellent combination properties in all of the CNF/RGO composite films based on the above experimental results; therefore, in order to further investigate its electrochemical performance, the CV at different scan rates and the GCD under changed current densities were detected. In addition, the stability of the electrode was tested. As shown in Figure 7d, the CV curves of CNFs/RGO_12_ present a nearly rectangle symmetrical shape when the charging rate changes from 10 to 100 mV s^−1^ and the voltage reversal current of potential at both ends responds quickly; this fact indicates that the CNFs/RGO_12_ film has good electrochemical capacitance behavior. Moreover, the characteristic of approximate symmetry for all of the GCD curves and the nonexistent voltage drop prove that it has good double-layer performance and high conductivity, as shown in Figure 7e.

The stability of the CNFs/RGO_12_ film electrode was estimated by measuring its galvanostatic charge/discharge from 0 to 1.0 V in 2000 cycles. The results are shown in Figure 7f, where the specific capacitance firstly increases and then drops during the 2000 cycles. The increase of the specific capacitance is attributed to the electrical activation of the electrode. After undergoing 2000 cycles, the initial retention rate of the specific capacitance is about 83.7%, which displays good cycling stability for the CNFs/RGO_12_ film. In addition, as a flexible conductive material, the effect of number of bends on capacitance was investigated by bending the CNFs/RGO12 film 200 times with a constant bending radius of 4 mm [60], and the measured results indicate that its capacitance did not change much, which is shown in Figure 7f.

## 4. Conclusions

In this work, self-supported flexible CNF/RGO composite films were fabricated using a combination of CNFs and GO through the processes of dispersion, vacuum filtration, gelation, and low-temperature drying. With this facile methodology, a compact three-dimensional network structure composed of CNFs and GO was gradually developed with a decreasing amount of water during vacuum filtration. Then, the gelation further strengthened the network of films due to the formation of the intramolecular and intermolecular hydrogen bonds of CNFs; therefore, the obtained CNF/RGO films exhibited excellent mechanical properties. On the other hand, the electrochemical performance of the CNF/RGO films was enhanced because the addition of TEMPO-oxidized CNFs restrained the restacking of graphene nanosheets and provided a great number of electrolyte channels during the reduction and low-temperature drying. These facts were completely proven by investigation using FESEM, FT-IR, XPS, Raman spectra, XRD, mechanical testing, and electrochemical performance measurements. For all CNF/RGO composite films, the CNFs/RGO_12_ film displayed outstanding comprehensive properties, for example, its tensile strength and conductivity were up to 83 MPa and 202.94 S/m, respectively, and its C_A_ value was as high as 146 mF cm^−2^ under a current density of 5 mA cm^−2^. Moreover, its capacitance did not change much after perpendicular bending 200 times. Therefore, the films obtained in this study offer possibilities and great potential for electrode materials as flexible energy storage devices.

## Figures and Tables

**Figure 1 molecules-25-02793-f001:**
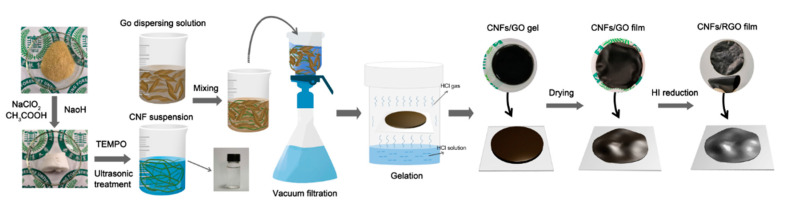
Schematic illustration of the fabrication process and structure of the cellulose nanofibril (CNF)/reduced graphene oxide (RGO) composite films.

**Figure 2 molecules-25-02793-f002:**
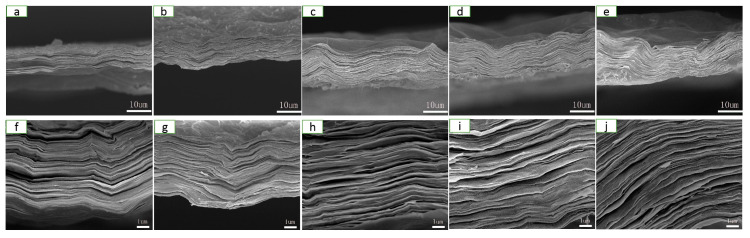
Cross-section field emission scanning electron microscope (FESEM) images of the (**a**,**f**) RGO, (**b**,**g**) CNFs/RGO_13_, (**c**,**h**) CNFs/RGO_12_, (**d**,**i**) CNFs/RGO_11_, and (**e**,**j**) CNFs/RGO_21_ films.

**Figure 3 molecules-25-02793-f003:**
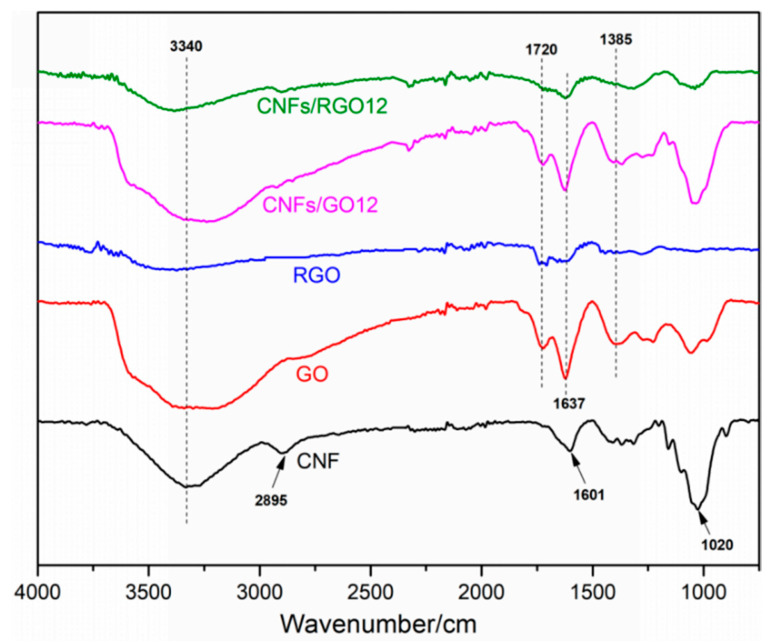
The FT-IR spectra of the CNF, GO, RGO, CNFs/GO_12_, and CNFs/RGO_12_ films.

**Figure 4 molecules-25-02793-f004:**
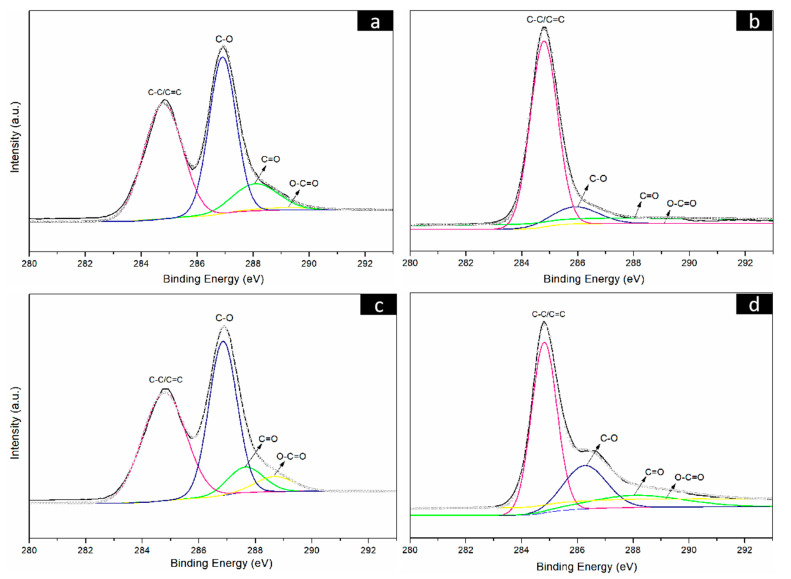
C1s spectra of the GO (**a**), RGO (**b**), CNFs/GO_12_ (**c**) and CNFs/RGO_12_ (**d**) films.

**Figure 5 molecules-25-02793-f005:**
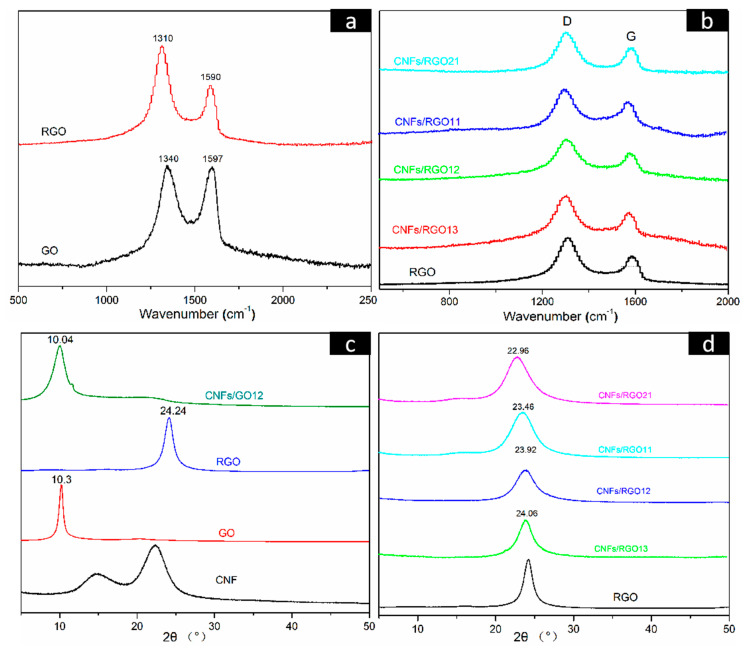
(**a**,**b**) Raman spectra of the GO, RGO, and CNF/RGO films. (**c**,**d**) X-ray Diffraction (XRD) patterns of the CNF, GO, RGO, CNFs/GO_12_, and CNF/RGO films.

**Figure 6 molecules-25-02793-f006:**
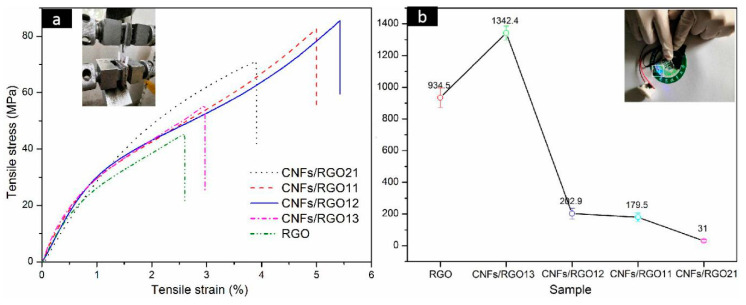
(**a**) Tensile stress−strain curves of the RGO and CNF/RGO films. (**b**) Conductivity of the RGO and CNF/RGO films.

**Figure 7 molecules-25-02793-f007:**
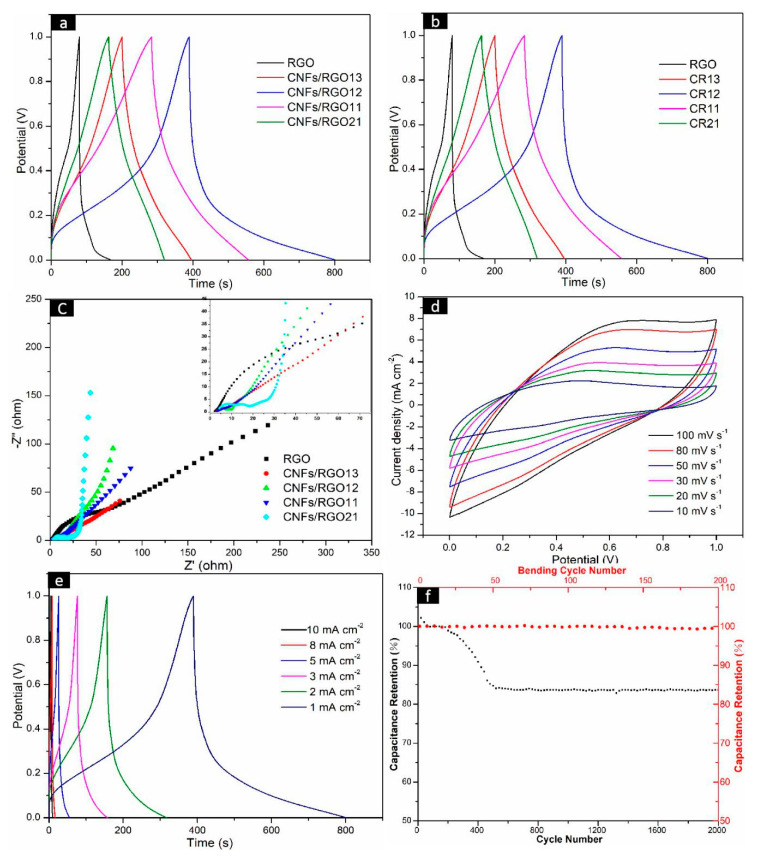
Electrochemical performance of the RGO and CNF/RGO composite film electrodes in a liquid electrolyte aqueous solution of 1 M H_2_SO_4_. (**a**) The galvanostatic charge/discharge (GCD) curves of the RGO and CNF/RGO films at 5 mV s^−1^. (**b**) Areal capacitance of the RGO and CNF/RGO film electrodes at different current densities. (**c**) The Nyquist impedance plots of the RGO and CNF/RGO film electrodes. (**d**–**f**) Cyclic voltammetry (CV) curves, GCD curves, and stability of the CNFs/RGO_12_ film electrode.

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
