# Peer review of "Fabrication and Performance of Self-Supported Flexible Cellulose Nanofibrils/Reduced Graphene Oxide Supercapacitor Electrode Materials"

_molecules, 2020, doi:10.3390/molecules25122793_

Round 1
Reviewer 1 Report
This paper reports composite films of cellulose nanofibrils and reduced graphene oxide, properties of the composite films, including tensile strength, electrical conductivity and energy storage as supercapacitor electrodes, have been tested. Overall, this is a relatively good work that can be published after major revision.
(1) Page 2, line 67: please check the logic. It is correct that high mechanical property is beneficial, but binder is actually added to the electrode to improve the mechanical property.
(2) Page 2, line 73: thickness of 5 mm is not correct, according to the SEM images.
(3) The reported electrochemical performance was measured using a three-electrode configuration, and current collector was not used, which could be the reason that an ideal capacitive behaviour with rectangular shaped CVs, and symmetric triangular shaped charge-discharge profiles was not achieved. I suggest that a two-electrode supercapacitor device with current collector to be assembled and tested. The author could refer to the literature for guidance: J. Power Sources 2014, 271, 269; Electrochim. Acta 2015, 182, 861.
(4) Figure 2, the scale bars are not clear.
(5) The interpretation of Raman results has to be improved, particularly for the evolution of Raman ID/IG ratio of GO after reduction. For guidance, please refer to literature: J. Am. Chem. Soc. 2017, 139, 17446.
(6) The gravimetric specific capacitance (F g-1) is a very important figure of merit for supercapacitor electrode material, therefore should also reported in the present work in addition to the areal capacitance.
(7) Page 9, line 302: an ideal capacitor would show a vertically straight line at low frequencies.
(8) Page 10, line 316, “etlectrostatic charge/discharge” should be galvanostatic charge/discharge.
(9) The cycling stability could be improved if using a two-electrode supercapacitor device for testing.
(10) What does perpendicular bending mean? The bending radius for the bending test should be provided; smaller bending radius would cause more damage to the film in one bending cycle. More information about bending radius can be found in literature: ACS Applied Materials & Interfaces 2019, 11, 32225.
(11) Language should be improved
Author Response
Response
Dear Mr. Luka Males and Reviewers:
Thanks very much for your comments concerning our manuscript entitled " Fabrication and performance of self-supported flexible cellulose nanofibrils/reduced graphene oxide supercapacitor electrode materials" (Manuscript ID: molecules-810625). Those comments are really valuable and helpful for us to revise and improve our paper. At the same time, it also has extremely important guiding significance for our researches. We have carefully studied the comments and made corrections which we hope meet with approval. All corrections in the paper and the responds to the reviewers’ comments are as follows:
Response to Reviewer #1:
- Page 2, line 67: please check the logic. It is correct that high mechanical property is beneficial, but binder is actually added to the electrode to improve the mechanical properties.
Response: After consideration, we have changed “Usually, the excellent self-supported flexible electrode materials require high mechanical properties and capacitance retention, therefore, the binder and conductive agents have to be removed in order to cut down weight itself” at page 2, line 64 to “However, excellent self-supporting flexible electrode materials require higher mechanical properties and capacitance retention, therefore, it is necessary to deal with the problem of bonding between materials”. Marked with yellow shading.
- Page 2, line 73: thickness of 5 mm is not correct, according to the SEM images.
Response: In page 2, line 71, “the thickness of 5 mm” refers to the thickness of the state before gelation, not the final sample. The original manuscript is ambiguous, so we have changed “the obtained mixtures were filtrated into film with a thickness of 5 mm under vacuum pressure, and then the film was placed in an acidic atmosphere to form hydrogel” to “the obtained mixture was filtered to a gelatinous thickness of 5 mm under vacuum pressure, and then it was placed in an acidic atmosphere to form a more complete hydrogel.” Marked with yellow shading.
- The reported electrochemical performance was measured using a three-electrode configuration, and current collector was not used, which could be the reason that an ideal capacitive behaviour with rectangular shaped CVs, and symmetric triangular shaped charge-discharge profiles was not achieved. I suggest that a two-electrode supercapacitor device with current collector to be assembled and tested. The author could refer to the literature for guidance: J. Power Sources 2014, 271, 269; Electrochim. Acta 2015, 182, 861.
Response: The reviewer's suggestions are very useful, let us assemble the device for testing to obtain better data. However, this research was to prepare a self-supporting flexible supercapacitor electrode material, we considered that as a self-supporting electrode material, the current test method using a three-electrode system has been able to characterize its performance well. But we will consider using this method in subsequent research, hoping to be understood.
- Figure 2, the scale bars are not clear.
Response: We processed the scale bars in Figure 2 to make it clearer.
- The interpretation of Raman results has to be improved, particularly for the evolution of Raman ID/IG ratio of GO after reduction. For guidance, please refer to literature: Am. Chem. Soc. 2017, 139, 17446.
Response: Raman analysis suggestions given by reviewers are very useful for this article. We have carefully studied “Two-Step Electrochemical Intercalation and Oxidation of Graphite for the Mass Production of Graphene Oxide”, and made some changes to the interpretation of Raman results. We have changed “this change is because a large number sp2 carbon atoms have been hybridized, which resulted in more disordered regions in the RGO nanosheets” to “this change confirmed that the partial restoration of the graphene lattice structure after reduction”, also, we quoted this article “J. Am. Chem. Soc. 2017, 139, 17446”, all changes marked with yellow shading.
- The gravimetric specific capacitance (F g-1) is a very important figure of merit for supercapacitor electrode material, therefore should also reported in the present work in addition to the areal capacitance.
Response: The purpose of this article is to prepare a self-supporting flexible composite film electrode. Although the gravimetric specific capacitance is very important for supercapacitor electrode materials, we believe that it is not necessary data in this study. At the same time, we are currently unable to conduct experiments for some reasons, so we may not be able to supplement the gravimetric specific capacitance data, hoping your understand.
- Page 9, line 302: an ideal capacitor would show a vertically straight line at low frequencies.
Response: We changed “the impedance curve was nearly a straight line in a low-frequency range” to “ an ideal capacitor would show a vertically straight line at low frequencies” at page 9, line 302. It was marked with yellow shading.
- Page 10, line 316, “etlectrostatic charge/discharge” should be galvanostatic charge/discharge. Marked with yellow shading.
Response: We have changed the “etlectrostatic charge/discharge” to galvanostatic charge/discharge at page 10, line 307. Marked with yellow shading.
(9) The cycling stability could be improved if using a two-electrode supercapacitor device for testing.
Response: We have considered that the amount of modification of the article using the two-electrode supercapacitor device for testing is too large and the purpose of our own electrochemical test, we think that using the three-electrode test can already do this, so we don’t plan to modify it.
(10) What does perpendicular bending mean? The bending radius for the bending test should be provided; smaller bending radius would cause more damage to the film in one bending cycle. More information about bending radius can be found in literature: ACS Applied Materials & Interfaces 2019, 11, 32225.
Response: In the article, our statement about the bending experiment is ambiguous, therefore, we have refered to “Screen-Printing of a Highly Conductive Graphene Ink for FlexiblePrinted Electronics”, and changed “the effect of number of bends on capacitance was investigated by perpendicularly bending the CNFs/RGO12 film for 200 times” to “the effect of number of bends on capacitance was investigated by bending the CNFs/RGO12 film 200 times with a constant bending radius of 4 mm” at page 10, line 312. . All changes marked with yellow shading.
- Language should be improved
Response: We have revised English.
Reviewer 2 Report
The authors present a paper entitled “Fabrication and performance of self-supported flexible cellulose nanofibrils/reduced graphene oxide supercapacitor electrode materials”, in which they present a study on the development of a simple method to obtain excellent self-supported flexible electrode materials with high mechanical properties and outstanding electrochemical performance by combining CNFs and RGO. The paper presents interesting results and deserves to be published after several revisions, as follows.
English is very poor and needs to be revised
Page 2 line 47, the first time the acronym TEMPO is introduced, should be explained (as it is done in the experimental section).
A clearer images of the produced samples should be added (maybe enlarging the images of the samples reported in figure 1)
Figure 5a-b the x-axis scale should be the same (it is not fully clear why the curves are reported in two different graphs, 5a and 5b).
Figure 6. The captions is wrong (exchange between a and b)
Figure 6, conductivity. The decimals are meaningless: the authors should remove them (203 instead of 202.94)
Figure 7c, authors should enlarge the scale, to make the curves more visibles
Author Response
Dear Mr. Luka Males and Reviewers:
Thanks very much for your comments concerning our manuscript entitled " Fabrication and performance of self-supported flexible cellulose nanofibrils/reduced graphene oxide supercapacitor electrode materials" (Manuscript ID: molecules-810625). Those comments are really valuable and helpful for us to revise and improve our paper. At the same time, it also has extremely important guiding significance for our researches. We have carefully studied the comments and made corrections which we hope meet with approval. All corrections in the paper and the responds to the reviewers’ comments are as follows:
- English is very poor and needs to be revised
Response: We have revised English.
- Page 2 line 47, the first time the acronym TEMPO is introduced, should be explained (as it is done in the experimental section).
Response: We have explained the first time the acronym TEMPO is introduced at page 2 line 47. Marked with yellow shading.
(3) A clearer images of the produced samples should be added (maybe enlarging the images of the samples reported in figure 1)
Response: We have replaced the final sample picture in Figure 1 to make it clearer.
(4) Figure 5a-b the x-axis scale should be the same (it is not fully clear why the curves are reported in two different graphs, 5a and 5b).
Response: We have modified the x-axis scale in Figure 5a to 500-2000 to keep it consistent with Figure 5b.
(5) Figure 6. The captions is wrong (exchange between a and b)
Response: We have modified the title of Figure 6 and exchanged a and b. Marked with yellow shading.
(6) Figure 6, conductivity. The decimals are meaningless: the authors should remove them (203 instead of 202.94)
Response: We have modified the decimal of conductivity in Figure 6, 202.9 instead of 202.94 and 31instead of 30.95, and we have added two illustrations to Figures 6a and b.
(7) Figure 7c, authors should enlarge the scale, to make the curves more visibles
Response: We have enlarged the scale of Figure 7c, and adjusted the layout of Figure 7.
Round 2
Reviewer 1 Report
The manuscript has been revised by the authors with acceptable improvements. I now recommend the publication of this work.